# The Role of Acquisition Angle in Digital Breast Tomosynthesis: A Texture Analysis Study

**Alessandro Savini [1,\*], Giacomo Feliciani [1,\*], Michele Amadori [2], Stefano Rivetti [3], Marta Cremonesi [4], Francesco Cesarini [5], Tiziana Licciardello [1], Daniela Severi [2], Valentina Ravaglia [6], Alessandro Vagheggini [7], Anna Sarnelli [1] and Fabio Falcini [8]**

[1] Medical Physics Unit, Istituto Scientifico Romagnolo per lo Studio e la Cura dei Tumori (IRST) IRCCS, 47014 Meldola, Italy; tiziana.licciardello@irst.emr.it (T.L.); anna.sarnelli@irst.emr.it (A.S.)

[2] Department of Radiology, Morgagni-Pierantoni Hospital, 47121 Forlì, Italy; michele.amadori@auslromagna.it (M.A.); daniela.severi@irst.emr.it (D.S.)

[3] Medical Physics Unit, Sassuolo Hospital, 41049 Sassuolo, Italy; s.rivetti@ospedalesassuolo.it

[4] Radiation Research Unit, European Institute of Oncology, 20141 Milano (MI), Italy; marta.cremonesi@ieo.it

[5] Post-Graduate School in Medical Physics, University of Bologna, 40126 Bologna, Italy; cfrancescocesarini@gmail.com

[6] Medical Physics Unit, Morgagni-Pierantoni Hospital, 47121 Forlì, Italy; valentina.ravaglia@auslromagna.it

[7] Unit of Biostatistics and Clinical Trials, Istituto Scientifico Romagnolo per lo Studio e la Cura dei Tumori (IRST) IRCCS, 47014 Meldola, Italy; Alessandro.vagheggini@irst.emr.it

[8] Romagna Cancer Registry, Istituto Scientifico Romagnolo per lo Studio e la Cura dei Tumori (IRST) IRCCS, 47014 Meldola, Italy; fabio.falcini@irst.emr.it

\* Correspondence: alessandro.savini@irst.emr.it (A.S.); giacomo.feliciani@irst.emr.it (G.F.); Tel.: +39-3274730398 (G.F.)

**Abstract:** Background: Digital breast tomosynthesis (DBT) systems employ a sophisticated set of acquisition parameters to generate an image set, and the DBT acquisition angle is considered to be one of the most important parameters. The aim of this study was to use texture analysis to assess how the DBT acquisition angle might influence DBT images of breast parenchyma. Methods: Thirty-four patients were selected from a clinical study conducted at IRST Institute. Each patient underwent a dual DBT scan performed with Fujifilm Amulet Innovality (Fujifilm Corp, Tokyo, Japan) in standard (ST, angular range = 15°) and high-resolution (HR, angular range = 40°) modalities. Texture analysis was applied on the paired dataset using histogram-based features and gray level co-occurrence matrix (GLCM) features. Wilcoxon-signed rank and Pearson-rank tests were used to assess the statistical differences and correlations between extracted features. Results: The DBT acquisition angle did not affect histogram-based features, whereas there was a significant difference in five GLCM features ($p < 0.05$) between DBT images generated with 15° and 40° acquisition angles. Correlation analysis showed that two GLCM features were not correlated at a $p < 0.05$ significance level. Conclusions: DBT acquisition angle affects the textures extracted from DBT images and this dependence should be considered when establishing baselines for classifiers of malignant tissue. Furthermore, texture analysis could be proposed as a quantitative method for comparing and scoring the contrast of DBT images.

**Keywords:** medical imaging; radiomics; tomosynthesis; acquisition angle

## 1. Introduction

Texture analysis is a promising technique for extracting quantitative features from radiographic images to be correlated with clinical or pathological characteristics of a tissue [1,2]. In the 1990s, several

authors began applying texture analysis to mammographic images to identify and classify malignant masses or microcalcifications [3–7]. Other authors correlated texture features with BRCA1/BRCA2 mutations [8] or with breast cancer subtypes [9]. In a recent work by Li et al., texture analysis of digital mammographic images was deemed as an excellent tool for differential diagnosis of benign and malignant tumors, especially if combined with standard visual analysis by radiologists [10]. Digital breast tomosynthesis (DBT) is a relatively new technique compared to digital mammography (DM), and for this reason, studies on texture analysis applied to DBT are less widespread. Kontos et al. studied the correlation between texture features calculated from DBT images and risk factors such as breast percent density or Gail and Claus risk models [11,12]. They concluded that texture features extracted from DBT images are more favorably correlated with risk factors than those extracted from DM. However, the authors also observed that the gray level distribution of DBT images may be affected by several factors such as the image acquisition geometry [13] and the reconstruction algorithm, hence suggesting that further research is needed to analyze the variability of features according to these parameters. At IRST institute, clinical DBTs can be performed with two acquisition geometries that differ in terms of the acquisition angle (15° and 40°) and in-plane image resolution (0.15 and 0.10 mm/pixel). With the final aim of developing a computer-aided diagnosis (CAD) system based on texture features, we first needed to investigate whether different acquisition geometries had an impact on texture values. Several authors have demonstrated that the parameter that most influences DBT images is the DBT acquisition angle. In particular, Li et al. found that an increased acquisition angle positively correlated with improved z-resolution [14]. Goodsitt et al. observed that increasing the DBT acquisition angle improved the contrast-to-noise ratio of disk details, as well as the subjective scoring of image quality by multiple readers [15]. Other authors used a model-observer approach to study the detectability of small signals in simulated DBT datasets, reporting that the acquisition angle had a positive effect on the detectability of each signal size considered [16]. A model-observer approach was also used recently by Lee et al., who studied the signal detectability of simulated DBT patterns by varying several acquisition parameters including angle, reconstruction filter, and slice thickness [17]. However, all these studies were performed on phantoms or simulated DBT datasets. The aim of the present study was to use texture analysis to characterize the impact of the acquisition angle on DBT images. This research was conducted on real DBT images of breast parenchyma originating from a clinical paired dataset (i.e., a single patient underwent dual DBT scan with 15° and 40° acquisition angle). To the best of our knowledge, no similar studies have been published elsewhere.

## 2. Materials and Methods

We used a Fujifilm Amulet Innovality (Fujifilm Corp, Tokyo, Japan) DBT system, which has 2 imaging modalities, i.e., a "standard" (ST) modality wherein the X-ray tube angular range continuously spans a range of 15°, and a "high-resolution" (HR) modality wherein the angular range is 40°. The number of acquired projections was 15 for both modalities. The in-plane pixel size of DBT reconstructed images was 0.1 mm in the HR modality and 0.15 mm in the ST modality. The slice thickness was 1 mm for both modalities. The automatic exposure control (AEC) of the HR modality was routinely tuned to deliver a higher dose to the breast than that of the ST modality. In order to eliminate this bias, we set up a scanning protocol to equalize the dose for both ST and HR modalities. This was carried out by applying the exposure parameters (i.e., anode/filter, kVp, mAs) for the HR modality, determined by the AEC during the ST acquisition.

This study was conducted as part of a clinical trial called "Digital Breast Tomosynthesis in a screening population—Investigation on angular range and dose level acquisition" currently ongoing at our institute (Protocol Code: IRST174.13). The trial was approved by the IRST Ethics Committee and was conducted in accordance with the ethical standards laid down in the 1964 Declaration of Helsinki. All patients gave written informed consent before being scanned. The trial inclusion criteria were: age between 45 years and 74 years and BI-RADS ≥ 3 evaluated with DM from the regional screening program. Exclusion criteria were: pregnancy, participation in another clinical trial, or a

clinical diagnosis of breast cancer. Patients satisfying the study criteria were scheduled to undergo a dual DBT scan in accordance with the above-mentioned scanning protocol. Each DBT scan was accompanied by an ultrasound (US) scan. The present study was conducted on a subset of the enrolled patients. In particular, we selected 34 patients (for a total of 36 image sets) whose DBT and US imaging results were scored as negative by 2 independent radiologists (i.e., BI-RADS = 1 after DBT and US re-evaluation). A follow-up of least two years was available for the selected patients.

We considered the craniocaudal (CC) view for each DBT image set, focusing the analysis on 7 slices at the center of the imaging study (Figure 1). Our choice was guided by Kontos et al., who suggested insulating the fibro-glandular tissue for feature extraction as superficial layers (skin and subcutaneous fat) only act as anatomical noise and are not correlated with breast cancer risk [11]. This choice is particularly relevant for small-sized breasts, as the number of slices in a DBT set is a function of breast thickness. Furthermore, preliminary analysis showed that mean feature values were not influenced by increasing the number of considered slices in large-sized breasts (data not shown). In order to apply a coherent texture analysis and insulate the effect of the acquisition angle, an in-house program developed in MATLAB (MathWorks, Inc., Natick, MA, USA) was used to downsample the in-plane resolution of HR images to the in-plane resolution of ST images, i.e., from 0.1 to 0.15 mm/pixel. The downsampling was performed using nearest-neighbor interpolation to maintain the integrity of original data and avoiding generating fictitious pixel values [18–20]. A region of interest (ROI) of $256 \times 256$ pixels was placed behind the nipple (Figure 1). This combination of ROI size and position was previously reported to have the highest discrimination power with regard to texture performance in DM [8,21] and in DBT [12]. Furthermore, the ROI size is a trade-off between different needs. In fact, the ROI size used in the present work is defined to include the limited parenchymal area of small breasts and is assumed standard for all breasts to avoid potential biases in the features' calculation. Feature computation was performed for each ROI in the 2D mode with MaZda software package [22]. We considered first-order features based on the image gray levels histogram and second-order features based on the gray level co-occurrence matrix (GLCM) [1,23] (Table 1). GLCM features were extracted using first-order neighbors and 0°, 45°, 90° and 135° directions. Feature values extracted from each slice were averaged into a single value. The same averaging operation was performed for the feature values obtained along the different GLCM calculation directions (0°, 45°, 90° and 135°). With these averaging operations, a single value for each feature was associated with a single DBT image set. Descriptions and mathematical details of the relevant features for this study can be found in Supplementary Materials.

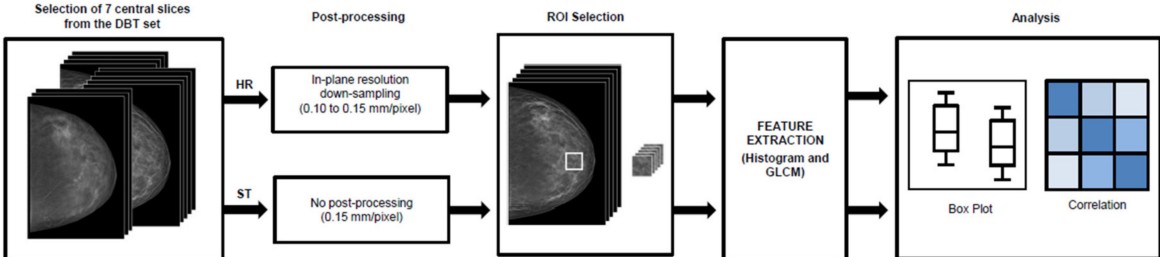

**Figure 1.** Workflow followed for texture analysis. DBT, digital breast tomosynthesis; ROI, region of interest; GLCM, gray level co-occurrence matrix.

In our DBT system, image gray levels had 14 bits and this resolution was maintained for histogram-based features. However, resampling was needed for GLCM features to avoid the risk of excessively large and sparse matrices that would inhibit a robust statistical analysis [24,25]. We extracted GLCM features after reducing the bits per pixel to 6 bits (MaZda default value), repeating the analysis with 5 bits and 7 bits to confirm the robustness of extracted features in relation to the bits rescaling.

**Table 1.** Features considered in this study. See [22] and Supplementary Materials for definitions.

| | |
|---|---|
| **First-Order Features (Histogram)** | Mean |
| | Variance |
| | Skewness |
| | Kurtosis |
| **Second-Order Features (GLCM)** | Angular Second Moment |
| | Contrast |
| | Correlation |
| | Sum of Squares |
| | Inverse Difference Moment |
| | Sum Average |
| | Sum Variance |
| | Sum Entropy |
| | Entropy |
| | Difference Variance |
| | Difference Entropy |

The features reported in Table 1 were extracted from the DBT images obtained with both ST (15° acquisition angle) and HR (40° acquisition angle) scan angles. All statistics comparisons were conducted considering these two paired datasets. Populations were compared with boxplots and the difference was assessed with a Wilcoxon-signed rank test (R-software version 3.5.3 Lucent Technologies, Murray Hill, New Providence, NJ, USA). A $p$-value $< 0.05$ was deemed statistically significant. Correlation between features that showed a significant difference between ST and HR images was inferred with Spearman rank correlation test. The datasets generated and/or analyzed during the current study are available from the corresponding authors on reasonable request.

## 3. Results

For a single DBT image set, feature values deviated within <1% from the value averaged over the slices and over the GLCM direction calculation. This low deviation justified taking the average value for each feature, corresponding to a single DBT image set. A total of 15 features were extracted and five of them (all belonging to the second-order group, i.e., GLCM features) showed significant differences between ST and HR images (see Table 2). The scenario was unchanged when the analysis was repeated after varying the bits' resampling amplitude (i.e., 5-bit, 6-bit and 7-bit). The correlation matrix of the features showing a significant difference between ST and HR images is reported in Figure 2. The Correlation feature was not correlated with any other feature. The Contrast feature was strongly correlated with other features (Spearman-$\rho = \pm 0.99$ with a $p$-value $< 0.001$) except the Correlation feature (Spearman-$\rho = 0.31$ with a $p$-value $= 0.14$). Again, this scenario was unchanged for 5-bit, 6-bit and 7-bit resampling amplitudes. For this reason, further analysis only took into consideration the not correlated GLCM features, Contrast and Correlation. Figure 3 shows boxplots comparing these features in ST and HR images. Figure 4 contains boxplots of Mean and Variance features that showed no significant difference between ST and HR images. Finally, in Figure 5, we show the differences between ST and HR in 2D space employing the two most relevant features.

**Table 2.** Differences between feature median values calculated from the high-resolution (40° acquisition angle) and standard (15° acquisition angle) DBT images. Only significant differences are reported ($p < 0.05$, Wilcoxon signed-rank test).

| | | % Difference 40°–15° Acquisition Angle ($p < 0.05$) |
|---|---|---|
| **First-Order Features (Histogram)** | | |
| | Contrast | +50% |
| | Correlation | −5% |
| **Second-Order Features (GLCM)** | Inverse Difference Moment | −14% |
| | Difference Variance | +45% |
| | Difference Entropy | +17% |

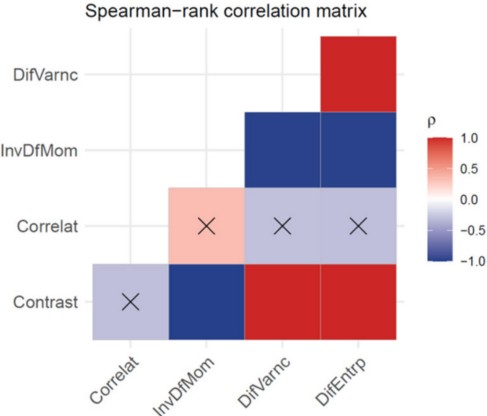

**Figure 2.** Spearman rank correlation matrix for features showing a significant difference between 15° acquisition angle (standard mode) and 40° acquisition angle (high resolution mode) DBT images. Red and Blue elements indicate a positive and negative correlation, respectively, with a significance: $p < 0.001$. Crossed elements indicate that the correlation is not significant $p > 0.05$.

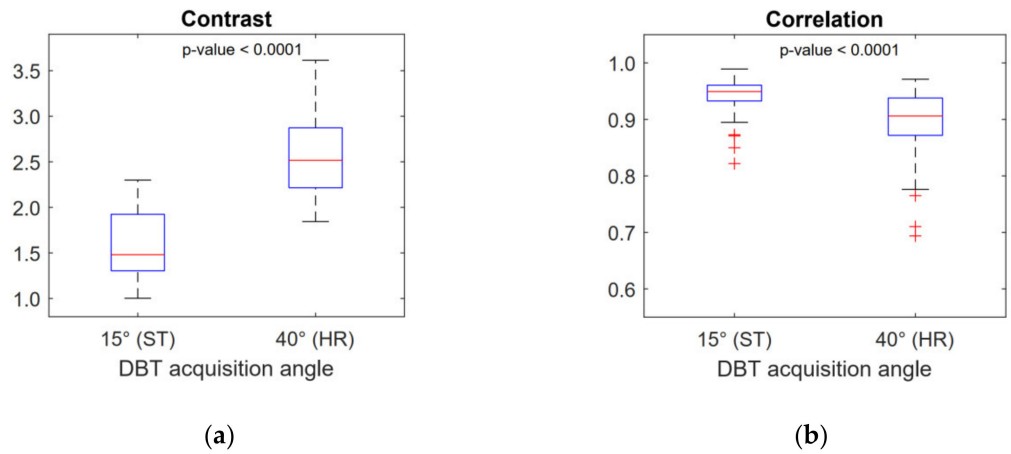

(**a**)　　　　　　　　　　　　　(**b**)

**Figure 3.** Boxplots of the non-correlated features Contrast (**a**) and Correlation (**b**). These features show a significant difference between 15° acquisition angle (standard mode) and 40° acquisition angle (high resolution mode) DBT images.

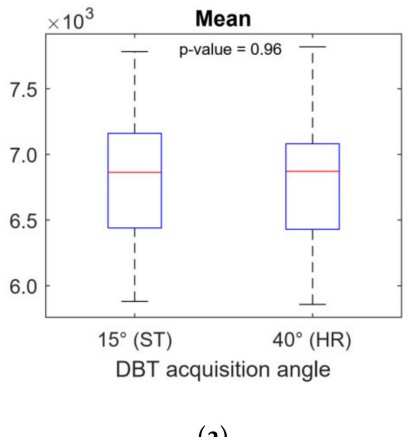

(**a**)

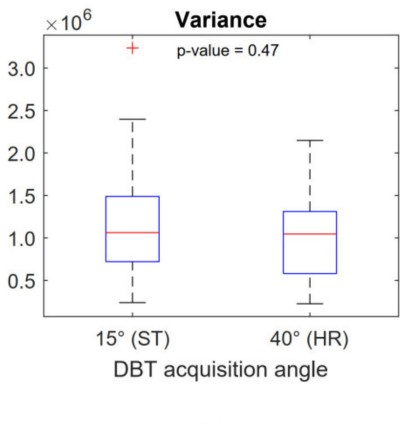

(**b**)

**Figure 4.** Boxplots of Mean (**a**) and Variance (**b**) histogram-based features. These features show no significant difference between 15° acquisition angle (standard mode) and 40° acquisition angle (high resolution mode) DBT images.

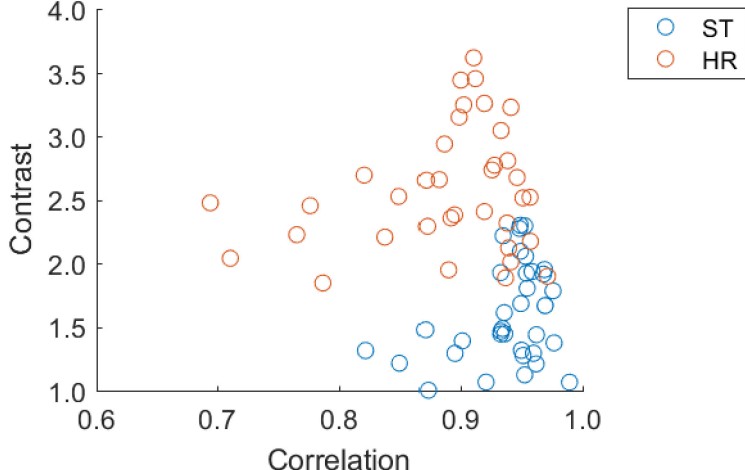

**Figure 5.** Scatter plot of Correlation vs. Contrast, the two most relevant features, showing the distribution in 2D space of the two imaging modalities.

## 4. Discussion

In the present work, we investigated the variability in texture features extracted from DBT of breast parenchyma in relation to the DBT acquisition angle. Insulating the effect of the acquisition angle was not a simple task. Different manufacturers use different angles, and comparisons are influenced by numerous confounding variables (e.g., different detector technology, different X-ray quality, different dose levels). The Fujifilm Amulet Innovality offers two acquisition modalities (ST and HR) on the same DBT system, which enabled us to eliminate many of these variables. We succeeded in insulating the DBT acquisition angle effect by designing a specific acquisition and data normalization protocol that equalized the image resolution and the dose of the two modalities. Furthermore, our comparison between feature values was performed on a paired dataset. The DBT acquisition angle influenced second-order features (from GLCM). GLCM features have been widely used in DM to build classifiers of malignant breast tissue [5,7,21,26,27]. Research has also been extended to DBT by other authors using different DBT systems with different acquisition angles [11,12,28]. In the present work, we demonstrated that the DBT acquisition angle significantly influences several GLCM features extracted from healthy breast parenchyma. This must be taken into consideration when developing classifiers for malignant tissue in DBT (i.e., the GLCM feature values for healthy breast tissue may differ simply because of the different DBT acquisition angle) and also when comparing texture values

calculated from different DBT systems, as these may differ on the basis of the acquisition angle. Texture features showing a significant difference may translate to physical considerations in DBT images and in the DBT acquisition process. In particular, the GLCM Contrast feature is a measure of the amount of local variation present in the image [23]. From this viewpoint, our results demonstrate that DBT images show a higher level of local contrast for healthy breast parenchyma (around 50% increase in median value) when DBT reconstruction is performed using projections with a wider acquisition angle (see Table 2). The observed difference can only be attributed to the wider acquisition angle because other variables such as higher dose or higher resolution were ruled out by the chosen acquisition protocol, and the image processing performed before the texture analysis. This result can be considered to be an addition to the existing findings regarding the quantitative comparisons between different DBT acquisition systems [29,30]. Our finding is coherent with findings of other authors regarding the positive effect of wider DBT acquisition angle on several quantitative parameters such as z-resolution, contrast-to-noise ratio of disk details, and subjective scoring of image quality by multiple readers [15,21]. In this context, texture analysis applied to DBT images has the potential to be an additional quantitative method for scoring and comparing DBT images.

Analogous considerations can be made regarding the difference observed for the Correlation feature, which is a measure of gray levels linear dependency in the image [23]. In our study, DBT images, reconstructed using wider acquisition angle projections, showed pixel gray levels which were less correlated.

Histogram-based features were not affected by DBT angular range, which may have been because histogram-based features do not depend on the local distribution of gray levels. Rather, they depend globally on the gray levels which were aggregated in the histogram within the ROI. For this reason, we could argue that local differences induced by different DBT acquisition angles were averaged out when looking globally at the histogram.

Our study had some limitations. We restricted the analysis to the features based on gray levels histogram and GLCM because these are the features most often considered by other authors for developing classifiers in DM [3–7] and DBT [11,12]. However, it might be useful in future studies to extend the analysis to other high-order features (e.g., run length matrix or wavelet features). In this study, we considered a dataset of 34 patients. The sample size was sufficient to reveal with statistical significance (i.e., $p < 0.05$) differences larger than 5% in DBT texture studies due to the different acquisition geometries. This fact needs to be considered when developing feature-based classifiers for healthy/malignant tissues. Statistical significance was obtained for 5 out of 15 considered features. Further studies will be performed with a larger dataset of patients in order to investigate subtler differences between the two imaging modalities.

The added value of our findings is that they were obtained by analyzing real images of breast parenchyma from a paired clinical DBT dataset. Kontos et al. [12] concluded that parenchymal texture features from DBT images have the potential to be used as imaging biomarkers to provide a more comprehensive quantitative characterization of breast parenchyma complexity. The texture analysis methodology applied in this study can be extended to assess the way in which the quantitative texture differences in DBT breast parenchyma might impact the clinical diagnostic performances.

## 5. Conclusions

In the present study, we found that texture features calculated from DBT images were influenced by the DBT acquisition angle, which is a key geometrical parameter of a DBT scan. This effect was observed for several GLCM-based features. Two main conclusions can be drawn from our results, the first being that the variability in texture values due to different acquisition geometries should be taken into consideration when developing GLCM texture-based classifiers for healthy/malignant tissue. Our second conclusion derives from the physical significance of some GLCM features. In particular, we demonstrated that the Contrast feature, defined from the GLCM, showed a significant increase (+50% in the median value) when a wider DBT acquisition angle was chosen. In this scenario, texture

analysis in DBT has the potential to be an alternative to other quantitative techniques such as model observer methods [16,17] for scoring and comparing DBT images. The approach used in this work could be extended to other clinical parameters and it could have the possibility of reducing the considerable effort of performing clinical studies with real human observers.

**Supplementary Materials:** The following are available online at http://www.mdpi.com/2076-3417/10/17/6047/s1, Feature mathematical description S1: Short tutorial about relevant features calculations and descriptions.

**Author Contributions:** Conceptualization, A.S. (Alessandro Savini), A.S. (Anna Sarnelli). and F.F.; methodology, G.F., V.R.; software, G.F., A.S. (Alessandro Savini), F.C.; validation, S.R., A.S. (Anna Sarnelli); formal analysis, G.F., A.S. (Alessandro Savini), A.S. (Anna Sarnelli), E.R.; investigation, M.A., F.C., D.S., F.F.; data curation, A.V.; writing—original draft preparation, A.S. (Alessandro Savini), A.S. (Anna Sarnelli), G.F.; writing—review and editing, A.S. (Alessandro Savini), G.F., A.S. (Anna Sarnelli), S.R., T.L., M.C.; visualization, A.S. (Alessandro Savini), G.F., V.R., T.L.; supervision, M.C., F.F. and A.S. (Anna Sarnelli); project administration, F.F.; All authors have read and agreed to the published version of the manuscript.

**Funding:** This research received no external funding.

**Acknowledgments:** The authors thank Gràinne Tierney for language editing.

**Conflicts of Interest:** The authors declare no conflict of interest.

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
