# Peer review of "The Role of Acquisition Angle in Digital Breast Tomosynthesis: A Texture Analysis Study"

_applsci, doi:10.3390/app10176047_

Round 1
Reviewer 1 Report
The paper analyzes one of the most important geometrical parameters involved when a medical image set is generated – the acquisition angle – and the way this parameter might influence the digital breast tomosynthesis (DBT) images. Their conclusion is that the acquisition angle affects the gray level co-occurrence matrix (GLCM) features extracted from DBT images. The comparison was made between the angles of 15 and 40 degrees. Not all of the references are very recent, but this is because texture features of medical images have been present in the researchers' concern for a long time.
The paper has a good general aspect, but more arguments and technical details are required:
- The study used 34 patients. Is this number enough to draw a reliable conclusion? How would be influenced the results if the number of patients would be significantly increased? The authors try to say something about this in lines 221-222 ("extending the sample size, subtler differences might arise" and "sample size was sufficient to..."), but it is not convincing enough.
- There were considered 7 slices from the center of an image set and the authors motivated their choice based on literature review. But there are still some questions: How many slices are generated for each patient? This number can be different from patient to patient. Is there any correlation between the chosen number of slices (7) and the total number?
- The region of interest (ROI) size and position were also set based on literature review. Maybe the authors can give some additional reasons.
- Table 1 presents the features considered in the study. The mathematical description of these features should also be provided. The authors say "See [22] for definitions", but the [22] reference is only a description of MaZda software package with text and nice images, but without any mathematical support.
- The authors say, in lines 30 and 62, "our Institute". Which one? There are eight affiliations, four of them involving two institutes.
The authors should also read again the paper and correct some errors:
- The word “Materials” from the beginning of line 82 seems to be useless there.
- In line 132, is that "è" before “[24,25]” necessary?
- The caption of Figure 2 is wrong (it belongs to Figure 1).
Author Response
We thank the Reviewer for all the precious comments and suggestions to improve the quality of our manuscript, we addressed all the questions point by point as follows.
Reviewer1:
The paper analyzes one of the most important geometrical parameters involved when a medical image set is generated – the acquisition angle – and the way this parameter might influence the digital breast tomosynthesis (DBT) images. Their conclusion is that the acquisition angle affects the gray level co-occurrence matrix (GLCM) features extracted from DBT images. The comparison was made between the angles of 15 and 40 degrees. Not all of the references are very recent, but this is because texture features of medical images have been present in the researchers' concern for a long time.
The paper has a good general aspect, but more arguments and technical details are required:
- The study used 34 patients. Is this number enough to draw a reliable conclusion? How would be influenced the results if the number of patients would be significantly increased? The authors try to say something about this in lines 221-222 ("extending the sample size, subtler differences might arise" and "sample size was sufficient to..."), but it is not convincing enough.
We thank the reviewer for this comment, we rephrased in Lines 245-250 in order to clarify our concepts as follows:
“In this study, we considered a dataset of 34 patients. The sample size was sufficient to reveal with statistical significance (i.e. p < 0.05) differences larger than 5% in DBT texture studies due to the different acquisition geometries. This fact needs to be considered when developing feature-based classifiers for healthy/malignant tissues. The statistical significance was obtained for 5 out of 15 considered features. Further studies will be performed with larger patients’ dataset in order to investigate subtler differences between the two imaging modalities.”
- There were considered 7 slices from the center of an image set and the authors motivated their choice based on literature review. But there are still some questions: How many slices are generated for each patient? This number can be different from patient to patient. Is there any correlation between the chosen number of slices (7) and the total number?
We thank the reviewer for pointing this out. The number of slices for patients is a function of breast size. However, we preferred to standardize our analysis by using 7 slices. In this way, we can account for small sized breasts and avoid low informative and potentially confusing superficial layers. Furthermore, preliminary analysis showed that mean features values were not influenced by increasing the number of slices over 7 but the results of this preliminary analysis are not shown in the present work for sake of brevity. We modified the text starting from line 111 as follows:
“This choice is particularly relevant for small sized breasts as the number of slices in a DBT set is function of breast thickness. Furthermore preliminary analysis showed that mean features values were not influenced by increasing the number of considered slices in large sized breasts (data not shown).”
- The region of interest (ROI) size and position were also set based on literature review. Maybe the authors can give some additional reasons.
Text in lines 122-125 has been modified to clarify that, beyond the literature review, the ROI size is a trade-off between different needs. In fact, the ROI size used in the present work is defined to include the limited parenchymal area of small breasts and is assumed standard for all breasts to avoid potential biases in the features’ calculation
- Table 1 presents the features considered in the study. The mathematical description of these features should also be provided. The authors say "See [22] for definitions", but the [22] reference is only a description of MaZda software package with text and nice images, but without any mathematical support.
We thank the reviewer for this suggestion and a small tutorial with feature equations and their description is now added as supplementary material
- The authors say, in lines 30 and 62, "our Institute". Which one? There are eight affiliations, four of them involving two institutes.
Lines were corrected specifying “IRST Insitute”.
The authors should also read again the paper and correct some errors:
- The word “Materials” from the beginning of line 82 seems to be useless there.
- In line 132, is that "è" before “[24,25]” necessary?
- The caption of Figure 2 is wrong (it belongs to Figure 1).
We thank the Reviewer for these corrections and apologize for the error in the caption of Figure 2, we fixed errors accordingly
Reviewer 2 Report
I have read this work several times and feel that it is publishable in the journal barred a few minor revisions. Largely, the only suggestion I have is that the tables be truncated to only contain relevant information. The main table of results had various factors in which no result was shown due to the fact that no significant difference was detected between conditions. Either show the result or remove the factor with this statement.
The graphs which are shown should have axes with the same number of decimal places throughout.
Other than these minor comments, I feel that this is a well designed and excellent study.
Author Response
We thank the Reviewer for all the precious comments and suggestions to improve the quality of our manuscript, we addressed all the questions point by point as follows.
Reviewer 2:
I have read this work several times and feel that it is publishable in the journal barred a few minor revisions. Largely, the only suggestion I have is that the tables be truncated to only contain relevant information.
- The main table of results had various factors in which no result was shown due to the fact that no significant difference was detected between conditions. Either show the result or remove the factor with this statement.
We thank the reviewer for this suggestion and we modified the table removing the not relevant features.
- The graphs which are shown should have axes with the same number of decimal places throughout.
We thank the reviewer for this suggestion and we modified the graphs accordingly
Other than these minor comments, I feel that this is a well designed and excellent study.
Reviewer 3 Report
The present work by Savini et al analyzes the impact of the acquisition angle in medical image tomography for breast cancer and on texture features. The paper is well written and easy to follow. The point is made clear from the beginning and the methods, results, and discussion are consistent. The message should be taken into account in studies using this kind of images. Overall it is a good paper.
Some additional and minor changes could be included:
- Features are a key element, however the reader is referred to reference [22]. Given the relevance of the features in this work, a short description of the features should be included, with emphasis on Entropy and Difference Entropy calculations, which sometimes bring bias in their implementations.
- Some further representation of features (in scatters for instance) could be informative for the interested reader, rather than or complementary to box plots.
- n the statistics sometimes it seems clear that paired statistic comparisons were used and sometimes it is not clear. Please revise and either use paired comparisons, or give a reason for not doing so, or make consistent.
Author Response
We thank the Reviewer for all the precious comments and suggestions to improve the quality of our manuscript, we addressed all the questions point by point as follows.
Reviewer 3:
The present work by Savini et al analyzes the impact of the acquisition angle in medical image tomography for breast cancer and on texture features. The paper is well written and easy to follow. The point is made clear from the beginning and the methods, results, and discussion are consistent. The message should be taken into account in studies using this kind of images. Overall it is a good paper.
Some additional and minor changes could be included:
- Features are a key element, however the reader is referred to reference [22]. Given the relevance of the features in this work, a short description of the features should be included, with emphasis on Entropy and Difference Entropy calculations, which sometimes bring bias in their implementations.
We thank the reviewer for this suggestion; a small tutorial with feature equations and their description is now added as supplementary material
- Some further representation of features (in scatters for instance) could be informative for the interested reader, rather than or complementary to box plots.
We thank the reviewer for this suggestion and we added a figure (Figure. 5) with a scatterplot of the two most relevant features in discriminating ST from HR.
- n the statistics sometimes it seems clear that paired statistic comparisons were used and sometimes it is not clear. Please revise and either use paired comparisons, or give a reason for not doing so, or make consistent.
We thank the reviewer for pointing this out and we added line 147: “All statistics comparisons were conducted considering paired datasets.”